# Artificial Intelligence in Higher Education: A Bibliometric Study on its Impact in the Scientific Literature

**Francisco-Javier Hinojo-Lucena, Inmaculada Aznar-Díaz** 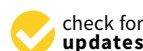**, María-Pilar Cáceres-Reche** **and José-María Romero-Rodríguez \***

Department of Didactics and School Organization, University of Granada, 18071 Granada, Spain; fhinojo@ugr.es (F.-J.H.-L.); iaznar@ugr.es (I.A.-D.); caceres@ugr.es (M.-P.C.-R.)

\* Correspondence: romejo@ugr.es; Tel.: +34-958-246-687

**Abstract:** Artificial intelligence has experienced major developments in recent years and represents an emerging technology that will revolutionize the ways in which human beings live. This technology is already being introduced in the field of higher education, although many teachers are unaware of its scope and, above all, of what it consists of. Therefore, the purpose of this paper was to analyse the scientific production on artificial intelligence in higher education indexed in Web of Science and Scopus databases during 2007–2017. A bespoke methodology of bibliometric studies was used in the most relevant databases in social science. The sample was composed of 132 papers in total. From the results obtained, it was observed that there is a worldwide interest in the topic and that the literature on this subject is just at an incipient stage. We conclude that, although artificial intelligence is a reality, the scientific production about its application in higher education has not been consolidated.

**Keywords:** artificial intelligence; emerging technologies; higher education; bibliometric study

## 1. Introduction

The advancement of technologies has resulted in a change of habits by much of the global population. People have modified the ways in which they connect, interact, read, write, and become informed through to the use of new technologies. In this scenario, the need arises for education to adapt to the current times and social customs. Therefore, the implementation of Information and Communication Technologies (ICT) in the classroom is a reality and should continue to be so. As indicated by Trujillo, López, and Pérez [1], digital literacy is more than justified in education, influencing the adaptation of university systems to the guidelines of The European Higher Education Area (EHEA). In this line, ICT in higher education represents a powerful resource that brings universities closer to the EHEA, since it advocates a methodological change in the teaching–learning process. In this way, the inclusion of technology allows for new dynamics of interaction in the classroom, in which transformative processes occur, leading to the implementation of learning methodologies focused on the student [2,3].

Reports of global relevance, such as the Horizon Report (a reference in educational technology), predict that artificial intelligence will be implemented in higher education within a period of four to five years [4]. Artificial intelligence is an emerging technology aimed at the creation of computational systems that present intelligent and adaptive behaviours, with the ability to learn from their environment, just like human beings [5,6].

Following León and Viña [7], artificial intelligence can contribute to changing education via the automation of administrative teaching tasks, software programs that favour personalized education,

the detection of topics that need reinforcement in class, the guidance and support of students outside the classroom, and the use of data in an intelligent way to teach and support the students [8].

In relation to the main systems on which artificial intelligence is applied in an educational context, we find, for example, intelligent tutors and intelligent teaching systems distributed over the internet [9,10]. In relation to intelligent tutors, they act as a guide to students' learning by detecting students' progress in learning based on the student's content knowledge and personal characteristics, while distributed intelligent teaching systems favour student collaboration through software programs that support and encourage interactions [11]. These studies highlighted three techniques of artificial intelligence in education: personalization systems (knowledge and individualized adaptation of the student), software agents (intelligent programs and robots with autonomy and the ability to learn) [12], and ontologies and semantic web (which gather knowledge from multiple spaces, Big Data) [13].

When developed and applied in education, these systems and techniques can be powerful resources for improving the teaching–learning process, since they are able to generate a kind of virtual teacher who is fully trained and has human characteristics, yet is able to interact ubiquitously (that is, at any time and place) [14].

In short, given the interest in the topic, it would be informative to search the scientific literature on artificial intelligence in higher education to understand the extent to which studies on this topic are present. Thus, an approximation was made to identify studies with the greatest impact. This search yielded interesting data both on the evolution of artificial intelligence in higher education over the years and on its future development.

Therefore, this paper analyzed the scientific production on artificial intelligence in higher education as indexed in Web of Science (WOS) and Scopus (2007–2017). The main motivation of the study is directly related to the purpose. Artificial intelligence is an emerging topic, and thus it is necessary for us to study and know its state of inclusion in the scientific literature. By doing so, we will be able to detect its scope and identify research trends for this emerging technology. This can help familiarize the readers with the topic and allow them to become more knowledgeable about the state of artificial intelligence in the scientific community. Likewise, the justification and significance of the analysis carried out within this study were based on three research questions that guided the work, which derived from the main motivation to know the state of artificial intelligence in the scientific literature and detect the source titles, organizations, authors, and countries with the highest scientific output on artificial intelligence in higher education:

-    What has the status of production been over time?
-    Is there a productive relationship between the number of authors and papers?
-    What are the main source titles, organisations, authors, and countries with the highest scientific output on artificial intelligence in higher education?

## 2. Methodology

In this paper, we used a methodology for bibliometric studies following the parameters set by the PRISMA declaration (Preferred Reporting Items for Systematic reviews and Meta-Analyses) [15]. For this, the focus of attention was placed on the metadata of the scientific production in the last 10 years (2007–2017), as collected in the databases with the greatest impact in social science: Web of Science (WOS) and Scopus [16].

Thus, first of all, the search equation "Artificial Intelligence" AND "Higher Education" was established, which was introduced in the WOS (Journal Citation Reports impact index—JCR) and Scopus (SCImago Journal & Country Rank impact index—SJR) databases. Afterwards, the results were narrowed down by scientific discipline (education, educational research) and years of production (2007–2017).

It should be noted that the keywords "artificial intelligence" and "higher education" are included in the ERIC Thesaurus as approved descriptors and used in Education Sciences. In the same way,

the union with the Boolean operator "AND" gives rigor to the search carried out [17]. The search took place during the last quarter of the year 2018.

*2.1. Sample*

The analysis was based on journal articles indexed in WOS and Scopus (*n* = 566). First, the study sample was established by the application of the inclusion criteria: education, educational research field, and year of publication (2007–2017). By applying the search equation in the two databases, a final study sample was obtained for each of them: WOS, *n* = 38, and Scopus, *n* = 94. The process to define the samples, with the application of the inclusion criteria is shown in the Figure 1.

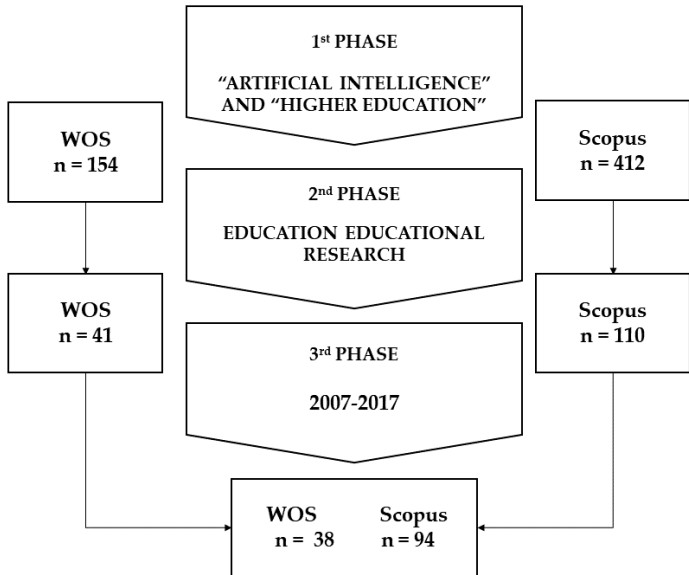

**Figure 1.** Phases of the bibliographic search process. WOS, Web of Science.

*2.2. Data analysis*

The analysis variables were established according to previous bibliometric studies [18–22]. Also, different bibliometric laws were applied: Price law and Lotka law [23–26]. Finally, different bibliometric indicators were established:

(1)    Output indicators: diachronic productivity and authors' productivity
(2)    Impact indicators: influence that some source titles, institutions, countries, and authors have on the scientific output on this topic

The data analysis was carried out on the basis of the information obtained in WOS and Scopus. In addition, some analyses required the use of the Excel Professional Plus 2013 software (Microsoft Corp., Redmond, WA, USA), for example, for correlations between authors and number of articles.

## 3. Results

*3.1. Output Indicators*

Considering the variable "year production", a similar evolution was observed in the first years in both databases. In contrast, a change was recorded in the year 2015: the production remained high in Scopus (18%), whereas it fell in WOS (5%). In 2017, the production in WOS increased and stabilized (16%), whereas in Scopus, it fell to the 2014 levels (12%). The results showed the maximum production peaks in each database, i.e., 18% in 2014 for WOS and 21% in 2016 for Scopus. Figure 2 shows the publication percentages on the topic artificial intelligence in higher education during the period 2007–2017.

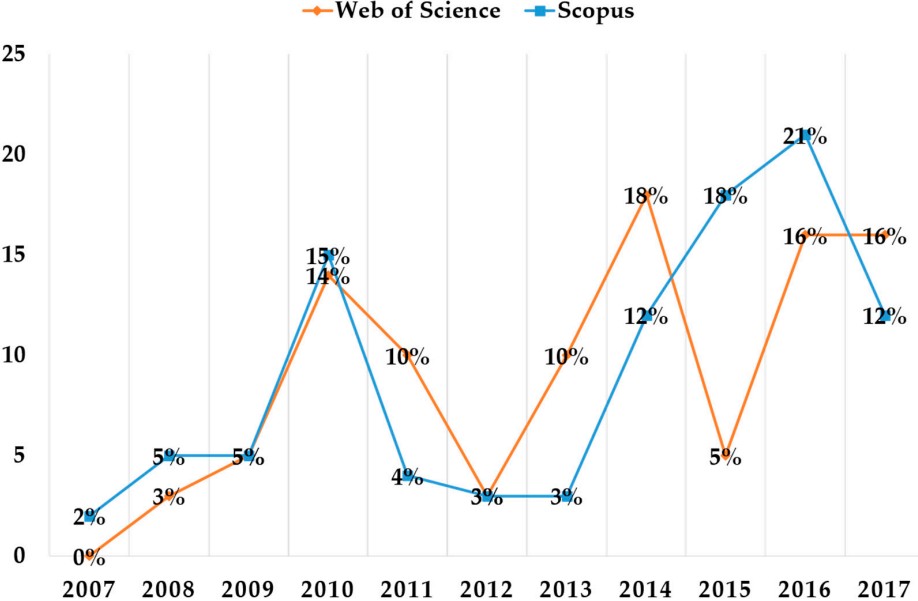

**Figure 2.** Scientific yearly production in WOS and Scopus.

The main premise of the Price law is that literature tends to duplicate after 10 years. This was confirmed in both databases, from 2007 (WOS 0%, Scopus 2%) to 2017 (WOS 12%, Scopus 16%). In addition, given the little growth of the scientific literature, it would be found in an incipient phase [19].

Nevertheless, authors' productivity is measured according to the number of published documents. Lotka's bibliometric law indicates that a small group of authors produce a large number of documents, these authors being very productive [21].

Our data confirmed Lotka's law. The graph in Figure 3 shows the negative correlation between the number of authors and the published papers, with the Pearson correlation coefficient $r = -0.922$ ($p$ value = 0.252). Therefore, few authors published the majority of articles. Besides, the model revealed a good calibration, explaining in 94% the variability of the y-axis in proportion to its average ($R^2 = 0.94$). In other words, the trend indicates a large number of documents corresponding to a small number of authors. However, despite the high value obtained in the Pearson correlation, the $p$-value is not significant.

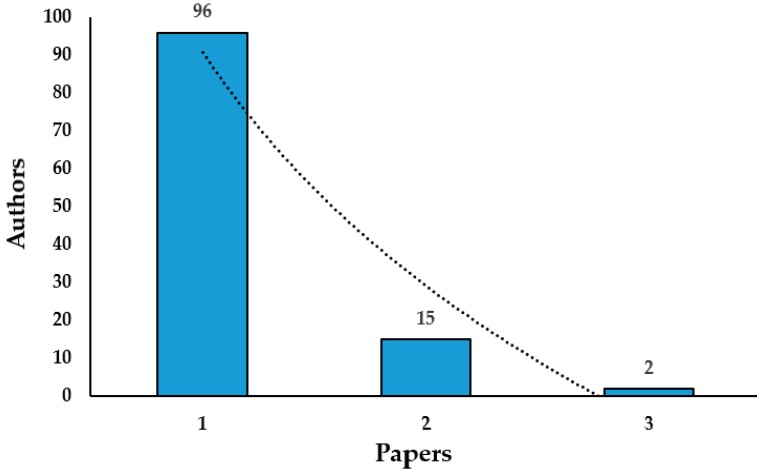

**Figure 3.** Correlation between authors (y-axis) and papers (x-axis).

*3.2. Impact Indicators*

Regarding the "document type", most of the production was concentrated in proceedings papers in both WOS 65.7% and Scopus 67%, followed by research articles, with 34.21% (WOS) and 27.65% (Scopus) of the total production.

This trend was found also when considering "source titles", which yielded more documents. Thus, in WOS, eight titles containing more than one reference accounted for 52.63% of the total scientific production. These are: *ICERI Proceedings* with four references (10.52%), *Edulearn Proceedings* with three references (7.89%), *Elearning and Software for Education* also with three references (7.89%), and the rest with two references each: *Asee Annual Conference Exposition* (5.26%), *Computers & Education* (5.26%), *Edulearn16 8th International Conference on Education and New Learning Technologies* (5.26%), *INTED Proceedings* (5.26%), and *On the Horizon* (5.26%).

In Scopus, the first 10 titles yielded 40.42% of the total production. These include: *Proceedings of International Conference On Artificial Intelligence And Education 2010 ICAIE* with eight references (8.5%), *Proceedings of The International Conference On E-Learning Icel* with five documents (5.3%), *IEEE Global Engineering Education Conference Educon* (4.25%) and *Proceedings of the European Conference On E-Learning Ecel* (4.25%) with four references each, *15th International Conference on Information Technology-Based Higher Education And Training Ithet* 2016 (3.19%), *Computer Applications in Engineering Education* (3.19%) and *Proceedings Frontiers in Education Conference Fie* (3.19%) with three references each. Finally, among the top ten with two documents each are *Computers in The Schools* (2.12%), *International Journal of Engineering Education* (2.12%), and *On the Horizon* (2.12%).

When we looked at the "organizations", the institutions with more references differed in the two databases. In addition, those that appeared in WOS were not found in Scopus. In WOS, these institutions were the University of Alicante (5.26%) and the University of California (5.26%) with two references, while in Scopus, the Firat University had three references (3.19%), and other institutions had two references as shown in Table 1.

**Table 1.** Scientific production of different Institutions as reported in WOS and Scopus.

| Organizations | WOS | | Scopus | | Citation | Impact Index |
|---|---|---|---|---|---|---|
| | n | % | n | % | | |
| University of Alicante (Spain) | 2 | 5.26 | - | - | - | - |
| University of California (USA) | 2 | 5.26 | - | - | - | - |
| Firat University (Turkey) | - | - | 3 | 3.19 | 1 | 0.33 |
| Polytechnic University of Valencia (Spain) | - | - | 2 | 2.12 | 4 | 2 |
| Aston University (UK) | - | - | 2 | 2.12 | 3 | 1.5 |
| Autonomous University of Madrid (Spain) | - | - | 2 | 2.12 | 6 | 3 |
| University of Ottawa (Canada) | - | - | 2 | 2.12 | 6 | 3 |
| University of Alcala (Spain) | - | - | 2 | 2.12 | - | - |
| Dublin City University (Ireland) | - | - | 2 | 2.12 | 2 | 1 |
| Hochschule Coburg (Germany) | - | - | 2 | 2.12 | - | - |
| University of Aveiro (Portugal) | - | - | 2 | 2.12 | 1 | 0.5 |

Note: Calculation of the Impact Index = Citations/Papers.

As for the "countries" (V5) that produced more literature, the United States were in first place in both databases, representing 23.68% (WOS) and 15.95% (Scopus) of the total production. However, the country in second place varied in the two databases, being Romania in WOS (15.78%) and Spain in Scopus (11.70%) (Table 2).

**Table 2.** Countries with highest scientific production in WOS and Scopus.

| Countries | WOS | | Scopus | | Citation | Impact Index |
|---|---|---|---|---|---|---|
| | n | % | n | % | | |
| Australia | 1 | 2.63 | 8 | 8.51 | 45 | 5 |
| Canada | 2 | 5.26 | 4 | 4.25 | 7 | 1.16 |
| China | 1 | 2.63 | 10 | 10.63 | 3 | 0.27 |
| Germany | 2 | 5.26 | 4 | 4.25 | 15 | 2.5 |
| Italy | 4 | 10.52 | - | - | - | - |
| Japan | 1 | 2.63 | 3 | 3.19 | 3 | 0.75 |
| Portugal | 2 | 5.26 | 6 | 6.38 | 9 | 1.12 |
| Romania | 6 | 15.78 | 2 | 2.12 | 7 | 0.87 |
| Spain | 4 | 10.52 | 11 | 11.70 | 16 | 1.06 |
| Turkey | 1 | 2.63 | 3 | 3.19 | 13 | 3.25 |
| United Kingdom | 3 | 7.89 | 7 | 7.44 | 25 | 2.5 |
| United States | 9 | 23.68 | 15 | 15.95 | 26 | 1.08 |

Note: Calculation of the Impact Index = Citations/Papers.

In relation to "authors", something similar as observed for "organizations" occurred, with differences in the two databases. Also, we considered authors with two or more references. In WOS, the most productive authors were Pertegal, ML, Jimeno, AM, Navarro, IJ, and Karakose, M. In contrast, in Scopus, the most productive author was Karakose, M with three references (3.19%) (Table 3).

**Table 3.** Authors with highest scientific production in WOS and Scopus.

| Authors | WOS | | Scopus | | Citation | Impact Index |
|---|---|---|---|---|---|---|
| | n | % | n | % | | |
| Pertegal, ML | 2 | 5.26 | - | - | - | - |
| Jimeno, AM | 2 | 5.26 | - | - | - | - |
| Navarro, IJ | 2 | 5.26 | - | - | - | - |
| Karakose, M | 2 | 5.26 | 3 | 3.19 | 1 | 0.2 |
| Akin, E | 1 | 2.63 | 2 | 2.12 | 1 | 0.33 |
| Ebert, M | - | - | 2 | 2.12 | 2 | 1 |
| Glynn, M | - | - | 2 | 2.12 | - | - |
| R-Moreno, MD | - | - | 2 | 2.12 | - | - |

Note: Calculation of the Impact Index = Citations/Papers.

Lastly, we obtained data relating the variable "the most-cited articles". This was chosen as a quality criterion to highlight studies with the highest impact that reported 10 or more citations. In both databases, this article was "Polite web-based intelligent tutors: Can they improve learning in classrooms?" published in 2011, with 24 citations in WOS and 28 citations in Scopus. We also observed that the first and second most cited articles were published in the journal *Computers & Education* (Table 4).

**Table 4.** The most cited articles in WOS and Scopus.

| Title | Authors | Journal | Year | WOS | Scopus |
|---|---|---|---|---|---|
| Polite web-based intelligent tutors: Can they improve learning in classrooms? [27] | McLaren, BM., DeLeeuw, KE., & Mayer, RE | Computers & Education | 2011 | 24 | 28 |
| Prediction of student's mood during an online test using formula-based and neural network-based method [28] | Moridis, CN., & Economides, AA | Computers & Education | 2009 | 20 | 24 |
| An enhanced Bayesian model to detect students' learning styles in Web-based courses [29] | García, P., Schiaffino, S., & Amandi, A | Journal of Computer Assisted Learning | 2007 | 17 | 21 |
| Development and validation of a learning analytics framework: Two case studies using support vector machines [30] | Ifenthaler, D., & Widanapathirana, C | Technology, Knowledge and Learning | 2014 | - | 30 |

## 4. Discussion and Conclusions

Artificial intelligence applied to higher education is a reality, since it is currently experimented, and beneficial results are being obtained [5,8,10,12,14]. However, at the same time, it is a marginal reality, since it is not developed enough, and its application is not widespread [2]. The data here reported show a boom of papers on artificial intelligence in recent years (2015–2016), although in Scopus, their number dropped in 2017 to the levels of 2014, and in WOS, it remained stable in 2016 and 2017. These observations have led to question its application in the period of four to five years indicated by the Horizon Report [4].

We found that most of the published documents type were proceedings papers, which highlights the interest in the topic which, however, does not seem enough to lead to the production of scientific articles. The gap between proceedings papers and scientific articles is still big: 65.7% of the former versus 34.21% of the latter in WOS, and 67% versus 27.65%, respectively, in Scopus. Likewise, *ICERI* and *ICAIE Proceedings* were the conferences with the highest number of documents on artificial intelligence in higher education (*ICERI*, 10.52% of the total production in WOS, and *ICAIE*, 8.5% of the total production in Scopus). In addition, the Price's law indicates that the scientific literature on this topic is in an incipient phase [23].

In confirmation of the Lotka's law, we found that few authors published two or more documents on artificial intelligence and higher education [25]. Most authors published only a paper on this topic.

On the contrary, when paying attention to institutions with two or more references, Spanish universities stood out (University of Alicante, Polytechnic University of Valencia, Autonomous University of Madrid, and University of Alcala) as the affiliations that produced more scientific literature (in total eight references in both databases, representing 5.26% in WOS, and 6.38% in Scopus, above the United States that had only two references in WOS (5.26%) from the University of California). In particular, the Autonomous University of Madrid obtained the highest impact index together with the University of Ottawa (Canada).

In relation to "countries", the United States is the country with the highest production, with 9 (WOS) and 15 (Scopus) documents, but the highest impact rate was obtained by Turkey, due to the number of citations in relation to the number of published documents. However, several countries of different continents (Europe, America, Asia, and Oceania) manifested their interest in the topic, since their production was at very high levels, placing them among the most productive countries. Thus, we conclude that this phenomenon is of global interest.

As for the authors, their heterogeneity was confirmed, since they were diverse, and most of them were not found in both databases. This indicates a distinction between authors who publish in journals indexed in WOS and those who publish in journals indexed in Scopus [16,17]. The author Akin, E was the greatest producer, with the highest impact index.

In conclusion, the most cited article on artificial intelligence in higher education in both databases [27] refers to the implementation of virtual tutoring as one of the main systems to the improvement of learning [9,10,14]. The second and third articles that appear in WOS and Scopus focus on intelligent systems to predict a student's mood ("Prediction of student's mood during an online test using formula-based and neural network-based methods") [28] and to detect learning styles ("An enhanced Bayesian model to detect students' learning styles in Web-based courses"), in line with the considerations on the applications of artificial intelligence in higher education [7,11].

In sum, this study has responded to the aim of analyzing the scientific production on artificial intelligence in higher education indexed in WOS and Scopus databases (2007–2017). We obtained relevant data to make inferences about its current status and its evolution in the coming years. Therefore, the data presented here show the evolution of artificial intelligence in higher education with a stagnation in the scientific production related to it in 2017, far from the considerations of reports recognized worldwide as the Horizon Report. In addition, the results provide the answers to the research questions that guided this paper: we established the state of the production of artificial intelligence in higher education over time, the relation between the number of authors and that of papers, and the main source titles, institutions, authors, and countries with the highest scientific output on artificial intelligence in higher education.

One methodological limitation of this bibliometric study is, typical of bibliometric studies in databases, is linked to the search engine. In fact, those articles that do not include the descriptors "Artificial Intelligence" and "Higher Education" in the title, summary, or keywords, can be excluded from the final results.

Finally, as a future research line, it would be interesting to continue improving the bibliometric analysis of the scientific production on artificial intelligence in higher education, pointing out the trends in the themes of publications about this topic for each year. Artificial intelligence applied to education must remain a focus of interest and attract more research, producing journal articles that will advance our knowledge of this topic and promote its real widespread application.

**Author Contributions:** Conceptualization, I.A.-D. and J.-M.R.-R.; methodology, M.-P.C.-R.; software, F.-J.H.-L.; formal analysis, J.-M.R.-R..; investigation, F.-J.H.-L., I.A.-D., M.-P.C.-R., and J.-M.R.-R.; writing—original draft preparation, I.A.-D. and J.-M.R.-R.; writing—review and editing, F.-J.H.-L., I.A.-D., M.-P.C.-R., and J.-M.R.-R.; visualization, M.-P.C.-R.; supervision, F.-J.H.-L.

**Funding:** This research received no external funding.

**Acknowledgments:** We acknowledge the researchers of the research group AREA (HUM-672), which belongs to the Ministry of Education and Science of the Junta de Andalucía and is based in the Department of Didactics and the School Organization of the Faculty of Education Sciences of the University of Granada.

**Conflicts of Interest:** The authors declare no conflict of interest.

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
