# Peer review of "Artificial Intelligence in Higher Education: A Bibliometric Study on its Impact in the Scientific Literature"

_education, doi:10.3390/educsci9010051_

Round 1

Reviewer 1 Report

The presented manuscript is focused in bibliometric study, the topic studied is 'artificial intelligence' connected with 'higher education'.

Single search query was used in two databases (WOS and Scopus), the results were presented in many ways.

In my opinion the text is interesting, but this is not scientific work, rather report, therefore I can not recommend it to publish in scientific journal.

From computer science point of view the presented analysis are simple.

The results from simple query are discussed.

There is no model created. The conclusions are not supported by scientific investigation.

No numerical experiments are performed.

The 38 papers from WoS and 94 papers from Scopus were analysed, therefore the precision is roughtly 0.01 (1%).

It is not corect to give results with bigger precision, as was done in Table 1, 2, 3, 4 and in text.

Author Response

Dear reviewer, 

I enclose the answers to your comments. Thank you in the first place for the effort made in attending this manuscript. We do not share the opinion that it is not a scientific work, since it is an article based on a bibliometric review methodology. This methodology is widely considered by the scientific community and there are several papers published in high impact journals in Journal Citation Reports, in the first quartiles. Examples of them:

1) Navarro, M.; Martín, M. Bibliometric Analysis of Research on Women and Advertising: Differences in Print and Audiovisual Media. Comunicar 2013, 41, 105-114.

2) Gutiérrez, C.; Martín, A.; Casasempere, A.; Fernández, A. Análisis Cientimétrico de la Grounded Theory en Educación. Revista de Educación 2015, 370, 121-148.

3) Heradio, R.; de la Torre, L.; Galan, D.; Cabrerizo, F.J.; Herrera-Viedma, E.; Dormido, S. Virtual and remote labs in education: A bibliometric analysis. Computers & Education 2016, 98, 14-38.

4) Hew, J.J. Hall of fame for mobile commerce and its applications: A bibliometric evaluation of a decade and a half (2000–2015). Telematics & Informatics 2017, 34, 43-66.

Reviewer 2 Report

In this paper that authors present a bibliometric study of the use of the words artificial intelligence and higher education in the scientific literature. The paper is largely a descriptive analysis of observations made around various metrics. In general, while the methodology appears sound, the lack of a clearly defined purpose of the study, or meaningful interpretation of the data seriously limits the value of the study. 

Major points 

1.    It is unclear exactly what issue (or issues) in AI the authors are attempting to address. They mention for example electronic tutoring and potential improvements through AI but the Abstract is vague and implies that the purpose of the study is to identify what AI in higher education looks like. However, the data presented and discussed is mostly around generic bibliometrics rather than the content of the papers. The authors should make the purpose of the study clearer. 

2.    Similarly, it is unclear what the authors are trying to claim are the main findings of the paper. This could perhaps be partly due to English language difficulties, but the final statement in the abstract, “it is found that although artificial intelligence is a reality, there is still a long way to go to fully establish itself in higher education.” is simply a generic and vague comment. Of course AI is a reality, and it is unclear what the authors mean by “fully establish itself in higher education”. Similarly, the final statement in the introduction including, “which yields interesting data” does nothing to tell the reader about the main findings of the paper. Why is the data interesting?

3.    The paper would benefit being read over by a native English speaker. There are numerous examples of poor English throughout the paper. While a reader can still get the sense of the paper, improving the English would certainly make it more readable. 

Author Response

Dear reviewer, 

I enclose the answers to your comments. Thank you in the first place for the effort made in attending this manuscript.

In attached file you will find the answers to your comments.

King regards,

Reviewer 3 Report

 The paper presents a bibliometric study on scientific production in artificial intelligence in higher education. Although the study and the presented numbers are interesting, the paper has several weaknesses. Namely:

- The motivation of the study is not clear. The authors should describe why they did the study and what is its expected contribution to the reader. Especially, the motivation of asking the research questions the authors ask should be argued.

- The indicators should be described in deeper details. For example, it is not mentioned, what the percentages in 3.1 and Figure 2 mean. The equation at the last line of page 5 should be explained (why logarithm?). How is R^2 defined? Correlation between authors and papers in Figure 3 should be also explained. The axes in graphs should be labeled. I will also appreciate adding tick marks on the axes of the graphs.

Some minor issues:

- The names of conferences/journal in section 3.2 should be highlighted (e.g. using italic font) for better readability.

-  A few grammar problems: “it is stands”, “it were applied”, “the WOS production it stabilizes”, “The results it show”, “In bot case”, “it highlight”, “we are find”, etc.

Author Response

Dear reviewer, thank you for your suggestions for improvement this paper. The answer to each of review comment is shown in the attached file.

Round 2

Reviewer 1 Report

I do not change my previous opinion. I do my review from computer science point of view. The single query database results without an model, without many numerical experiments, without error measure, should not be accepted to publication in scientific journals.

Author Response

Dear reviewer, two databases have been reviewed: Scopus and Web of Science, the two most important scientific databases in the social sciences.

We understand your position about bibliometric studies, but the reality is very different since they are considered scientific articles by the scientific community. Examples of this are the articles published in high impact journals in Journal Citation Reports, in the first quartiles.

1) Navarro, M.; Martín, M. Bibliometric Analysis of Research on Women and Advertising: Differences in Print and Audiovisual Media. Comunicar 201341, 105-114.

2) Gutiérrez, C.; Martín, A.; Casasempere, A.; Fernández, A. Análisis Cientimétrico de la Grounded Theory en Educación. Revista de Educación 2015, 370, 121-148.

3) Heradio, R.; de la Torre, L.; Galan, D.; Cabrerizo, F.J.; Herrera-Viedma, E.; Dormido, S. Virtual and remote labs in education: A bibliometric analysis. Computers & Education 2016, 98, 14-38.

4) Hew, J.J. Hall of fame for mobile commerce and its applications: A bibliometric evaluation of a decade and a half (2000–2015). Telematics & Informatics 2017, 34, 43-66.

Reviewer 3 Report

I still have objections to the graph in Fig. 3. There is no reason why the function is continuous, as its domain is only a discrete set {1,2,3}. A bar graph (histogram)  will be more appropriate.   

Author Response

We agree with the appreciation of the reviewer, it is more appropriate to reflect the graph with a histogram. Therefore, the figure has been modified according to this suggestion.
